# HIV-related stigma and uptake of antiretroviral treatment among incarcerated individuals living with HIV/AIDS in South African correctional settings: A mixed methods analysis

Lucy Chimoyi[1]*, Christopher J. Hoffmann[1,2], Harry Hausler[3], Pretty Ndini[1], Israel Rabothata[1], Danielle Daniels-Felix[3], Abraham J. Olivier[3], Katherine Fielding[4,5], Salome Charalambous[1,4], Candice M. Chetty-Makkan[1,4,6]

1 Implementation Research Division, The Aurum Institute, Johannesburg, South Africa, 2 Department of Medicine, Johns Hopkins University, Baltimore, Maryland, United States of America, 3 TB HIV Care, Cape Town, South Africa, 4 School of Public Health, University of the Witwatersrand, Johannesburg, South Africa, 5 Department of Infectious Disease Epidemiology, London School of Hygiene and Tropical Medicine, London, United Kingdom, 6 Health Economics and Epidemiology Research Office, Johannesburg, South Africa

* LChimoyi@auruminstitute.org

## Abstract

### Background

Stigma affects engagement with HIV healthcare services. We investigated the prevalence and experience of stigma among incarcerated people living with HIV (PLHIV) in selected South African correctional settings during roll-out of universal test and treat.

### Methods

A cross-sectional mixed-methods study design included 219 incarcerated PLHIV and 30 in-depth interviews were conducted with four different types of PLHIV. HIV-related stigma was assessed through survey self-reporting and during the interviews. A descriptive analysis of HIV-related stigma was presented, supplemented with a thematic analysis of the interview transcripts.

### Results

ART uptake was high (n = 198, 90.4%) and most reported HIV-related stigma (n = 192, 87.7%). The intersectional stigma occurring due to individual and structural stigma around provision of healthcare in these settings mostly contributed to perceived stigma through involuntary disclosure of HIV status. Interpersonal and intrapersonal factors led to negative coping behaviours. However, positive self-coping strategies and relationships with staff encouraged sustained engagement in care.

**Data Availability Statement:** All relevant data are within the paper and its Supporting information files.

**Funding:** The authors acknowledge the support of the U.K. Department for International Development (DFID)/UKAID under grant MMM/EHPDA/AURUM/05150013 to SC; IAVI and University of California, San Francisco's International Traineeships in AIDS Prevention Studies (ITAPS) award U.S. NIMH, R25MH0647, to Prof Krysia Lindan. The funders had no role in study design, data collection and analysis, decision to publish, or preparation of the manuscript.

**Competing interests:** The authors have declared that no competing interests exist.

## Conclusion

We encourage continuous peer support to reduce stigmatization of those infected with HIV and whose status may be disclosed inadvertently in the universal test and treat era.

## Introduction

HIV prevalence in South African correctional facilities is significantly higher than that of the general population and estimated to range between 9% and 41% [1,2]. The HIV risk in this population increases through needle sharing, violence, unprotected, sometimes non-consensual and unplanned sexual encounters, and limited condom negotiation [3–5]. The HIV treatment policy by the South African Department of Correctional Services (DCS) to improve treatment access was introduced to enable HIV control in this high-risk population [6]. The United Nations Office on Drugs and Crime (UNODC) and the World Health Organisation (WHO) also recommend delivery of pre-exposure prophylaxis (PrEP) for correctional facilities [7]. This recommendation is not yet adopted in these settings that focus more on upholding security [3].

The rollout of universal test and treat (UTT) in South Africa from 2016 appears to be the best strategy for facilitating antiretroviral treatment (ART) delivery and uptake, particularly in correctional settings where alternative prevention approaches such as pre-exposure prophylaxis have specific challenges to implementation [6,8]. Correctional facilities provide an opportunity for successfully engaging individuals in treatment and sustained retention in care after release into the community [9,10]. For expansion of delivery and uptake of HIV services, addressing known barriers, such as stigma, is necessary [11–14]. Individuals experience incarceration stigma which is further compounded by living with HIV [15–17]. As a result, they may experience enacted (overt acts of hostility and discrimination due to HIV status), perceived (anticipated discriminatory acts as a result of HIV status) or internalised (acceptance of negative beliefs and feelings about HIV) stigma [11]. The intersection of these types of stigma create barriers to engagement in HIV care services [17].

Existing evidence has highlighted the negative influence of HIV-related stigma on engagement along the HIV care continuum [18,19]. The different levels of HIV-related stigma are associated with undesirable health and social consequences [20]. Internalized stigma is linked to depressive symptoms [21] whereas perceived and enacted stigma heightens difficulties with disclosure, delays with treatment initiation and leads to poor treatment adherence [22].

Evidence on the effect of HIV-related stigma among incarcerated populations is available in developed countries [13,23]. Findings have identified structural and individual factors that negatively affect engagement in care including stigma and lack of social support from professional healthcare providers [13,15,16,24,25]. In South Africa, HIV-related stigma has been extensively examined in the general population [20,26,27], but not in incarcerated populations. Our study aimed to understand 1) the uptake of ART; 2) the prevalence of HIV-related stigma among ART users; and 3) the experience of HIV-related stigma among incarcerated people living with HIV within the implementation of UTT in these settings.

## Methods

### Study design

This analysis was nested within a parent study [*Treatment as prevention* (TasP)], that investigated the feasibility of implementing UTT in correctional facilities. We conducted a

mixed-methods study [28] among incarcerated people with HIV in selected South African correctional facilities. First, we analysed quantitative data to describe ART uptake and prevalence of HIV-related stigma among ART users. Then, we qualitatively explored the experiences of HIV-related stigma among incarcerated people living with HIV. Lastly, we integrated our quantitative and qualitative findings.

## Study setting

The TasP study was conducted between September 2016 and March 2018 in three correctional facilities: Johannesburg (Gauteng Province), Brandvlei and Worcester (Western Cape Province). The Johannesburg correctional centre (JCC) is a large facility situated in Gauteng, a high HIV burden province; with an estimated HIV prevalence of 25.3% [2]. Brandvlei (BCF) and Worcester (WCF) correctional facilities are located within the Breede Valley region in the Western Cape. HIV prevalence is estimated at 10.9% and 9.2% at each facility, respectively [2].

## Data collection for the quantitative component

The TasP study enrolled incarcerated participants living with HIV aged $\geq$18 years, in facility $\geq$ 3 months, not on ART and not anticipated to leave the facility within 6 months and followed-up for twelve months. At enrolment, trained research assistants collected baseline information from participants. Follow-up data were collected from clinic files at three time-points: one, six and twelve months after enrolment. At the six-month time-point, research assistants administered a psychosocial survey that assessed stigma, disclosure and social support (S1 File). Our study utilized data collected six months post-enrolment.

## Data collection for the qualitative component

From March 2017, trained research assistants approached and recruited study participants from the correctional facilities for qualitative in-depth interviews (IDIs). For inclusion, participants had to be $\geq$ 18 years, stayed in the facility for $\geq$ 6months, and agreed to audio recording of interview sessions. Recruitment was through purposive sampling using pre-specified criteria to include: HIV participants not engaged in care; consistently engaged in care with documented viral load suppression (<50 copies/ml); and those in care but with unsuppressed viral load (> 1,000 copies/ml) who were enrolled in the cohort study (Fig 1).

Trained male and female study staff with varying qualifications (Diploma, Bachelors, Masters and PhD Level) conducted IDIs in private rooms located within the facilities using a semi-structured interview guide (S2 File). At each interview, two study staff, one as the main interviewer and a second as the note taker, were present. The main interviewer administered the written informed consent, collected demographic information and conducted the interview. All personal identifiers were removed and participants were assigned a unique study number. The interviews, lasted between 45 and 90 minutes and were conducted in a language of the participants' preference that included English, isiZulu, isiXhosa, Afrikaans, Setswana and Sesotho. No remuneration was offered and transcripts were not returned to participants for comment. The probing technique was initially reviewed for the first five interviews by investigators and feedback provided to teams to adjust the exploration of themes in more detail. We reached saturation of themes during data collection by reviewing the transcripts, providing frequent feedback to research assistants on probes to use during the interviews and including participants whose experiences of HIV care services varied.

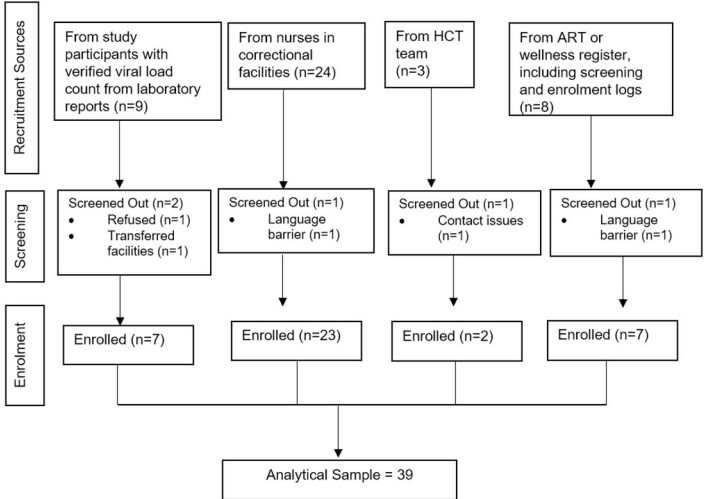

**Fig 1. Flow diagram of participants for in-depth interviews (n = 30) in the Treatment as Prevention qualitative sub-study from 31 March 2017–31 March 2018 in Gauteng and Western Cape provinces, South Africa.**

## Quantitative measurements

ART uptake was derived from the recorded date of ART initiation. Stigma was assessed using nine questions adapted from The People Living with HIV Stigma (PLHIV) Index and a score ≥1 was used to indicate overall stigma [20]. HIV-related stigma was assessed by combining the six HIV- related stigma questions. Four of these questions assessed HIV-related stigma, two focussed on reluctance to seek treatment and the last two on stigma associated with incarceration. Participants choose "Yes (1)", "No (0)", or "No answer (3)" and responses with "No answer" set to missing before combining. A scoring ≥1 indicated stigma as previously used and a numerical dichotomized variable with "Yes" and "No" categories was created.

Voluntary disclosure was assessed quantitatively and qualitatively by asking participants whether they disclosed their HIV- positive status to family, spouse/partner, friend, fellow inmate, work colleagues or corrections services staff member.

## Qualitative measurements

From qualitative interviews, any mention of negative attitude or action towards a participant based on HIV positive status by peers or corrections staff was coded as HIV-related stigma. Experiences with HIV related stigma were described to reflect social (experienced) and outcome expectations (perceived). Disclosure was explored by asking participants whether they had willingly revealed their HIV- positive status to family, spouse/partner, friend, incarcerated individual (inmate), work colleague or corrections services staff member.

## Data analysis and management of the quantitative component

Categorical data were summarized as frequencies and percentages (proportions) while continuous data were summarized as medians, and interquartile ranges (IQR). We did not assess for differences in socio-demographic, stigma measures and psychosocial factors by ART status. Data analysis was conducted using Stata version 14 [29].

## Data analysis and management of the qualitative component

Research assistants transcribed all interviews verbatim and back translated to English audio recordings with renderings of local languages. Co-authors (IR and PN), fluent in either study language checked the accuracy of the transcripts against digital recordings. LC reviewed the process notes during the analysis and authors resolved all queries prior to analysis. Once verified, transcripts were imported into NVivo version 10 for coding and analysis [30].

To analyse the qualitative data, we used thematic analysis. Two members of the research team (LC and CMC-M) independently reviewed and coded the transcripts. Transcripts were read multiple times for familiarity with the data, patterns identified, and initial codes generated. We adapted a previously developed framework for HIV-Related Stigma, Engagement in Care, and Health Outcomes to explore the intersection between individual and structural stigma [31] in an incarcerated population. Codes were organized into four themes that affected engagement in care, namely structural factors, HIV-related stigma, individual factors (interpersonal and intrapersonal), structural factors (correctional environment, staff attitude and involuntary disclosure) and self-coping mechanisms (Fig 2). Collaboratively, researchers discussed and revised emerging themes and sub-themes in an iterative and inductive manner. During analysis, further refining and development of codes and categories was done until we reached a saturation point where no new themes were evident. For instance, a theme on structural factors was created from perceived stigma to include the structural challenges that inadvertently disclose HIV status such as public pill lines, special diet delivery, or ART clinic days. Ambiguities and disagreements on assigning the themes and coding for instance internalized stigma were resolved by discussion. Final themes were summarized, and described using direct quotes.

## Ethics statement

This study protocol was jointly approved by the University of the Witwatersrand human research ethics committee (Wits HREC; Number 150510) and Department of Correctional Services (DCS) research ethics committee. Ethics approval was granted to use direct quotes and written informed consent was obtained from participants. No special privilege was offered for participation and their participation would not influence decisions from the parole boards. All participant records and information were anonymized and de-identified prior to analysis.

## Results

### Quantitative findings

Two hundred and nineteen participants, mostly male (156, 71.2%), median age 34 years (IQR: 29–39) completed a psychosocial survey at 6 months post-enrolment (Table 1). Most participants were single (n = 153, 69.9%), had never been incarcerated before (n = 142, 64.8%), spent less than one year in the facility (n = 129, 58.9%), and were incarcerated at the Gauteng province facility (n = 166, 75.8%). At six months 95.9% (n = 210) reported stigma and 87.7% (n = 192) HIV-related stigma. Most participants voluntarily disclosed their HIV status (n = 207, 94.5%) and many reported receiving social support from friends and family (n = 171, 78.1%). Out of 219 participants, 198 (90.4%) had initiated ART by 6 months. Of those who reported no HIV-related stigma (n = 27), 22 (81.5%) started ART and of the 192 who reported HIV-related stigma, 176 (91.7%) started ART. Of those comfortable with HIV status, 128/137 (93.4%) had started ART whereas of those not comfortable with HIV status, 56/67 (83.6%) had started ART (Table 2). Of those reluctant to access ART 33/40 (82.5%) had started ART and of those not reluctant to access ART, 157/170 (92.4%) had started ART. Of those who thought it

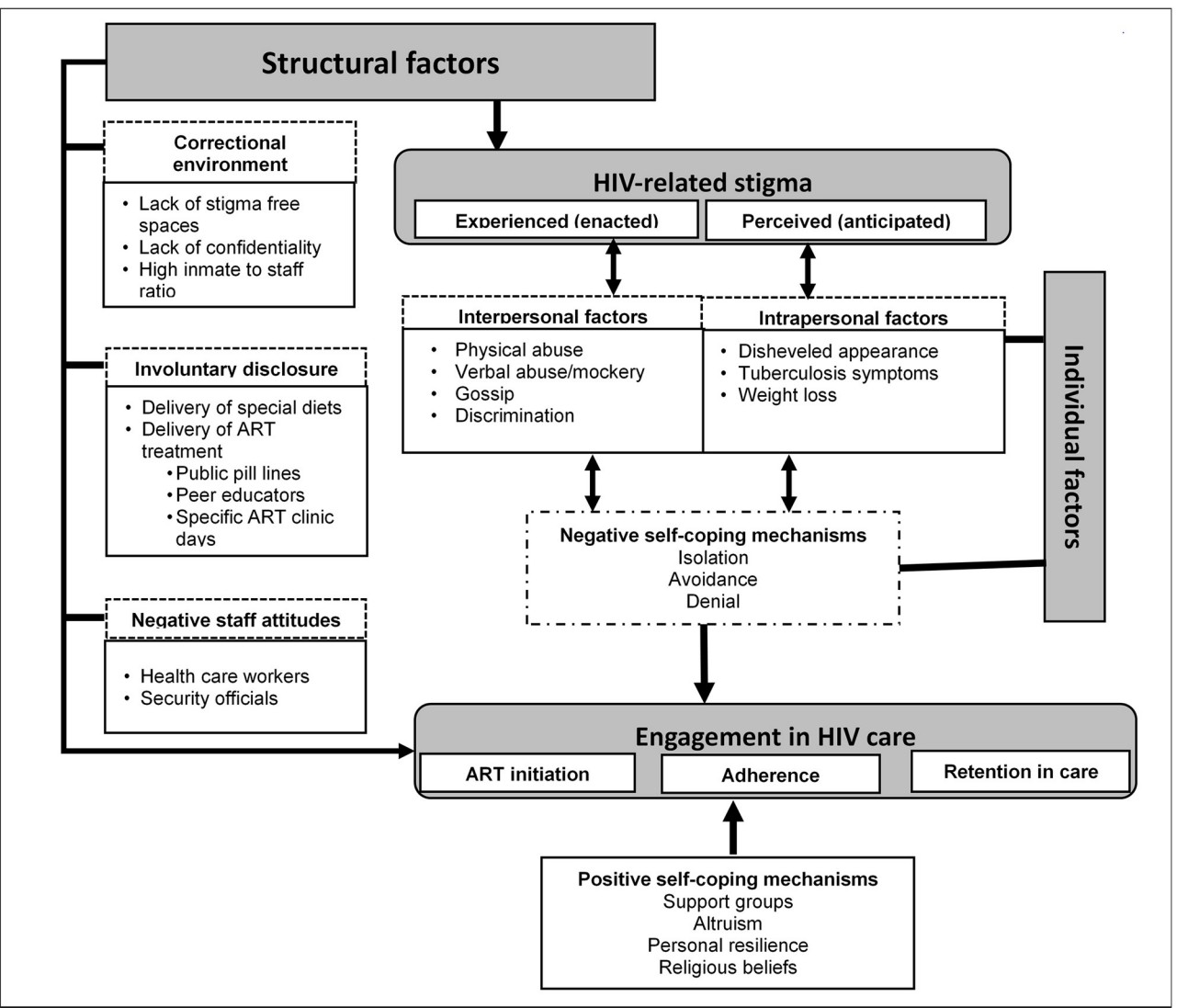

**Fig 2. Structural and individual factors contributing to HIV-related stigma that affect engagement in care in correctional settings.**

was important to keep HIV status a secret 83/87 (95.4%) had started ART and from those who do not think it is important to keep HIV status a secret, 105/120 (87.5%) had started ART.

## Qualitative findings

Fig 1 describes the recruitment sources of 30 IDI participants, of median age 35 (IQR: 29–38) years. Of these, 27 (69%) were single, and had spent a median number of 4 years incarcerated (IQR: 1–8). During inductive reasoning when describing attributes of gender and sites, there were no distinctive differences across and within themes. Table 3 outlines the main themes and sub-themes were developed around the existing structural factors (involuntary disclosure, lack of stigma free spaces and negative staff attitudes), HIV-related stigma (perceived and experienced) and individual interpersonal and intrapersonal factors. Another developing theme was self-coping mechanisms (negative and positive).

**Table 1. Socio-demographic characteristics of TasP study participants who completed the psychosocial survey 6-months post-enrolment (N = 219) at selected correctional facilities in, South Africa (2016–2018) by ART uptake.**

| Characteristics | Total (N = 219) | | Not on ART (n = 21) | | On ART (n = 198) | | p-value |
|---|---|---|---|---|---|---|---|
| | n | (%)[†] | n | (%)[†] | n | (%)[†] | |
| **Demographic characteristics** | | | | | | | |
| **Age group (years)** | | | | | | | |
| <25 years | 20 | (9.1) | 2 | (10.0) | 18 | (90.0) | |
| ≥25 years | 199 | (90.9) | 19 | (9.5) | 180 | (90.5) | 0.60 |
| **Gender** | | | | | | | |
| Male | 156 | (71.2) | 13 | (8.3) | 143 | (91.7) | |
| Female | 63 | (28.8) | 8 | (12.7) | 55 | (87.3) | 0.32 |
| **Marital status** | | | | | | | |
| Currently Married | 55 | (25.1) | 4 | (7.3) | 51 | (92.8) | |
| Previously Married | 11 | (5.0) | 3 | (27.3) | 8 | (72.7) | |
| Never married | 153 | (69.9) | 14 | (9.2) | 139 | (90.8) | 0.15 |
| **Previously incarcerated** | | | | | | | |
| No | 142 | (64.8) | 16 | (11.3) | 126 | (88.7) | |
| Yes | 77 | (35.2) | 5 | (6.5) | 72 | (93.5) | 0.34 |
| **Time incarceration (months)** | | | | | | | |
| <12 months | 129 | (58.9) | 12 | (9.3) | 117 | (90.7) | |
| ≥12 months | 90 | (41.1) | 9 | (10.0) | 81 | (90.0) | 0.86 |
| **Site (province)** | | | | | | | |
| Gauteng | 166 | (75.8) | 20 | (12.0) | 146 | (88.0) | |
| Western Cape | 53 | (24.0) | 1 | (1.9) | 52 | (98.1) | 0.03 |
| **Psychosocial characteristics** | | | | | | | |
| **Reported any stigma** | | | | | | | |
| No | 9 | (4.1) | 1 | (11.1) | 8 | (88.9) | |
| Yes | 210 | (95.9) | 20 | (9.5) | 190 | (90.5) | 0.64 |
| **Reported any HIV-related stigma[*]** | | | | | | | |
| No | 27 | (12.3) | 5 | (18.5) | 22 | (81.5) | |
| Yes | 192 | (87.7) | 16 | (8.3) | 176 | (91.7) | 0.09 |
| **HIV disclosure** | | | | | | | |
| No | 12 | (5.5) | 2 | (16.7) | 10 | (83.3) | |
| Yes | 207 | (94.5) | 19 | (9.2) | 188 | (90.8) | 0.32 |
| **Received social support** | | | | | | | |
| No | 48 | (21.9) | 3 | (6.3) | 45 | (93.8) | |
| Yes | 171 | (78.1) | 18 | (10.5) | 153 | (89.5) | 0.58 |

TasP: Treatment as Prevention; IQR: Interquartile range; TB: Tuberculosis; ART: Antiretroviral treatment;

[†]Percentages reported as row%;

[*] derived from six questions PLHIV stigma index.

## Fear of inadvertent HIV positive status disclosure while accessing care in correctional facilities

Due to the lack of privacy in a congregate setting fear of HIV status disclosure was reported by majority of the participants. Specific procedures such as use of public pill lines (lining or queuing for medication), special diets, distribution of medication by peers and being escorted to the clinic on specific ART days further increased disclosure of HIV status. Some patients

**Table 2. Reported experiences of stigma from TasP study participants by ART status at selected correctional facilities in Gauteng and Western Cape Provinces, South Africa (2016–2018).**

| Individual stigma measures | | Not on ART (n = 21) | | On ART (n = 198) | | p-value |
|---|---|---|---|---|---|---|
| | | n | (%) | n | (%) | |
| Ashamed of their HIV status* (n = 210) | No (n = 160) | 13 | (8.1) | 147 | (91.9) | 0.22 |
| | Yes (n = 50) | 7 | (14.0) | 43 | (86.0) | |
| Comfortable with HIV status while incarcerated compared to in community* (n = 204) | No (n = 67) | 11 | (16.4) | 56 | (83.6) | 0.03 |
| | Yes (n = 137) | 9 | (6.6) | 128 | (93.4) | |
| Important to keep HIV status a secret while incarcerated* (n = 207) | No (n = 120) | 15 | (12.5) | 105 | (87.5) | 0.05 |
| | Yes (n = 87) | 4 | (4.6) | 83 | (95.4) | |
| Lost respect or standing in corrections because of HIV status* (n = 208) | No (n = 178) | 18 | (10.1) | 160 | (89.9) | 0.32 |
| | Yes (n = 30) | 1 | (3.3) | 29 | (96.7) | |
| Possible to keep HIV status a secret while incarcerated* (n = 210) | No (n = 111) | 12 | (10.8) | 99 | (89.2) | 0.50 |
| | Yes (n = 99) | 8 | (8.1) | 91 | (91.6) | |
| Ashamed to access health care in correctional facilities (n = 210) | No (n = 185) | 19 | (9.7) | 166 | (85.1) | 0.48 |
| | Yes (n = 25) | 1 | (4.0) | 24 | (96.0) | |
| Reluctance to access ARVs in correctional facilities* (n = 210) | No (n = 170) | 13 | (7.6) | 157 | (92.4) | 0.06 |
| | Yes (n = 40) | 7 | (17.5) | 33 | (82.5) | |
| Ashamed of being in the correctional facilities (n = 210) | No (n = 69) | 4 | (5.8) | 65 | (94.2) | 0.22 |
| | Yes (n = 141) | 16 | (11.3) | 125 | (88.7) | |
| Lost respect or standing in community because of incarceration (n = 209) | No (n = 80) | 8 | (10.0) | 72 | (90.0) | 0.87 |
| | Yes (n = 129) | 12 | (9.3) | 117 | (90.7) | |

*The six HIV-stigma related questions.

**Table 3. Number of participants reporting specific themes related to HIV-related stigma in selected correctional facilities in Gauteng and Western Cape Provinces, South Africa (2016–2018).**

| Main Theme | Sub-themes | Number of participants |
|---|---|---|
| Fear of inadvertent HIV positive status disclosure while accessing care in correctional facilities | Involuntary disclosure | 25 |
| | Lack of stigma free spaces | 18 |
| | Negative staff attitudes | 15 |
| Perceived (anticipated) stigma and engagement in care | Anticipated prejudice | 12 |
| | Anticipated rejection of services | 8 |
| Experienced (enacted) stigma and lack of privacy and confidentiality | Verbal abuse | 12 |
| | Physical abuse | 1 |
| | Social exclusion | 7 |
| | Gossiping | 19 |
| Negative self-coping mechanisms and disengagement from care | Denial of HIV positive status | 3 |
| | Isolation from peers | 6 |
| Positive self-coping mechanisms and engagement in care | Personal resilience | 9 |
| | Religious beliefs | 18 |
| | Social (Peer) support systems | 10 |
| | Altruism | 10 |

received an extra portion of food and collected their meals on trays mostly referred to as "*lap-tops*" outside the dining hall instead of queueing in order to receive a special diet.

> *"The people that are on ARVs their laptops come already dished out and they don't go and queue in the kitchen. So, the trolley comes in with the food it stands outside and then the people go and actually fetch their food".*
>
> *Male, 33 years, PLHIV.*

To alleviate the disclosure concerns from incarcerated PLHIV, some staff from the correctional facility took steps to minimize disclosure. For example, some healthcare workers sought to provide care in private.

> *"You then wait but when we are given our pills we go in one by one, and then you are given. When they give you the pills, they make sure that the door is closed and if there is someone who does not take the same medication as yours, he is made to wait outside so that you are not given pills in front of him. If the pills are the same, they are able to give them to you at the same time."*
>
> *Male, 32 years, PLHIV.*

**Perceived (anticipated) stigma and engagement in care.**   Some participants reported awareness of stigma leading to non-engagement for other inmates. Participants specifically reported that stigma deterred their HIV care engagement.

> *"Well, there is obviously stigma here (within correctional facilities) as much as it is outside (the correctional facility). You will hear it from comments and will discourage one from actually starting treatment, to be honest".*
>
> *Male, unknown age, PLHIV.*

Due to the anticipated stigma, participants rejected services from the peer supporters who assisted healthcare workers in providing care by distributing pre-packaged medication. Most incarcerated PLHIV reported this mode of delivery as one that breached confidentiality of HIV status.

> *"At the beginning, i said there is no confidentiality. The nurses' assistants (referring to peer educators) shout out loud during clinic visits that those queuing for the tablets (ART) must step forward to collect. Now they are saying that when I am sick that I should report to the assistant! Where is the confidentiality? Isn't the assistant a monitor?"*
>
> *Female, 40 years, PLHIV.*

**Experienced (enacted) stigma and lack of privacy and confidentiality.**   Participants from all facilities reported experiencing stigma through discrimination, gossip, or physical and verbal abuse. Discrimination was a feeling of social exclusion by peers due to HIV status such as rejection and isolation.

*"I was rejected to the extent that I could not stay with other inmates. And couldn't share a bath with other inmates of which we were staying with a single cell."*

*Female, 23 years, PLHIV.*

The theme on violence emerged from all facilities and across genders. Most described physical abuse which was not related to HIV status. In our study, there was a single report of violence associated with HIV status.

*"When I came to the sentenced yard, a few months later, another girl also shouted me about my status and then the two of us also fought."*

*Female, 23 years, PLHIV.*

Verbal abuse was mentioned and was at times related to the HIV status of an individual. This manifested as insults and mockery with derogatory words meted out on incarcerated PLHIV as this participant reported:

*"There is a guy who lived in our room; he is one guy who really abused me because each time he talked about this illness of mine. So he was evicted from the room".*

*Male, 36 years, PLHIV.*

Verbal abuse through negative comments from correctional staff occurred in all facilities. This potentially affected the relationship between corrections staff and incarcerated HIV positive individuals although its effect on health seeking behaviour was uncertain as explained below.

*"Some are good and understanding about HIV status, but some are rude. One official once insulted me and I should have retaliated but then again I just respected his position. That is why I prefer to stay on my bed because I am trying to avoid problems".*

*Male, 34 years, PLHIV.*

Gossiping from staff or fellow inmates, defined as talking negatively about HIV-positive individuals mostly occurred after inadvertent or deliberate disclosure of one's status. Participants mentioned having feelings of shame when peers talked about them. They often felt enraged; enough to resort to physical abuse such as explained by one participant.

*"Yes, there were two girls talking about my status, one of the unsentenced (girls) shouted at me about my status and then we fought. . .then they moved her. When I came to the sentenced yard, a few months later, another girl also shouted me about my status and then the two of us also fought."*

*Female, 23 years, PLHIV.*

**Negative self-coping mechanisms and disengagement from care.** Negative self-coping strategies were reported as denial of status and/or isolation through experiences of stigma.

Due to the inaccurate HIV beliefs from peers, some incarcerated PLHIV were anxious about using communal facilities for fear of victimization leading some to report social distancing. As one participant reported:

*"At that point I didn't have knowledge about HIV, so I was thinking all of those things that the people that I'm living with, are they going to accept me as I am? 'Will they mind sharing a cup with me knowing that I'm HIV positive?' 'Will they mind sharing a toilet with me, knowing that I'm HIV positive? Isolating myself from a lot of people because I know that I am HIV positive, you understand instead of being around them".*

*Male, 34 years, PLHIV.*

**Positive self-coping mechanisms and engagement in care.** Incarcerated PLHIV showed positive self-coping mechanisms, for instance personal resilience by investing in improving one's health and that of others, after becoming comfortable with their status while incarcerated. The need to uphold a healthy status was important especially for male participants who would continue with the leadership role in their families upon release as reported in this extract.

*"So, there are kids that are younger than me that are looking up to me. I want to go outside get a job maybe. So, that's why I get motivated to take my treatment so I don't get sick."*

*Male, unknown age, PLHIV.*

Some participants cited anchoring their faith in religious beliefs encouraging acceptance of HIV status and engagement in care.

*"I thought to myself that no I do not have a choice you see being in HIV positive and in prison. I have to accept the road that I travel, you see. First, I am a Christian. Not a playing Christian. I do not play at church. I worship. I am holding on to Jesus. Only God knows and so you will not complain when you are with Him, you see".*

*Male, 33 years, PLHIV.*

Some participants formed informal support groups that provided social support and counselling. This facilitated adherence to medication at a specified time despite the side effects and stigma by talking openly about HIV.

*"I do not find any challenges, because most of us, maybe 99.9% here in prison, all are HIV positive. Yes. I am open about my status. We are free to talk, especially here in our cell. We remind each other to take our medication. The ladies say, you will never die when I'm still here"*

*Female, 33 years, PLHIV.*

Altruism emerged when participants reported helping fellow incarcerated PLHIV peers cope with their experiences and urging them to stay in care as expressed below.

*"Even if you can give him treatment, but you need to be there for him. Because really? Maybe it is important for him to start immediately, but you need to be there for him. Or, you need to get some support somehow. Because remember this person never scheduled his life to getting this treatment the way it happens here",*

*Male, 36 years, PLHIV.*

## Discussion

In this study, we described HIV-related stigma and care engagement in correctional facilities. Our findings confirm the existence of stigma in correctional settings for incarcerated people living with HIV; and the fear of involuntary disclosure while seeking HIV care. Participants in this study demonstrated high uptake of antiretroviral treatment (92%) despite a high reported prevalence of HIV-related stigma. Our findings also suggest that in this population the adoption of positive self-coping mechanisms and a positive relationship between the staff contributed to their sustained HIV care engagement.

Concerns around the consequences of inadvertent HIV status disclosure in correctional settings have been widely documented [13,25,32,33]. There was fear among participants that the correctional services environment neither promoted confidentiality of HIV diagnosis nor allowed for privacy when receiving HIV services. Similar to community and clinic settings, the consequences for disclosure of HIV status in this setting could lead to enacted stigma and may have affected engagement in care [15]. Perhaps, the consequences of disclosure and impact of stigma among incarcerated populations is less severe than in non-correctional settings without the associated risks of partner abandonment, loss of employment or social standing in the community [34]. This may also explain the high proportion of PLHIV that were comfortable with their status while incarcerated.

The study illuminated the lack of confidentiality when accessing HIV care in the correctional facilities. Similar to previous findings, our study highlighted how privacy concerns during distribution of HIV care and treatment affected health service utilization by incarcerated PLHIV inmates [35–37]. The distribution of HIV treatment on specific days by nurses or peer educators at ART clinics and/or patient queueing [17,38] are modes of delivery of care revealed by inmates as having little regard for privacy. Consequently, some participants reported avoiding engagement in care due to a fear of disclosure and anticipated stigma [15]. In some cases, incarcerated PLHIV reported attempts to conceal medication from peers [24] or shun engagement in care [38]. As reported by Shalihu et al (2014), the provision of special meals with additional protein-rich foods for PLHIV initiating HIV treatment in Namibia exposed inmates to unintended disclosure and potential stigmatization [16]. Despite this being a standard practice across different settings, the loss of confidentiality from the delivery of these meals in correctional facilities may increase enacted HIV-related stigma [38].

Our study findings highlighted intersectional stigma occurring as a result of the convergence of being incarcerated while living with HIV. Our findings showed the negative consequences of individual and structural stigma on incarcerated PLHIV which worsened health seeking behaviours and outcomes [34]. Denial of HIV positive status and social exclusion from healthcare services and peers were reported to decrease HIV disclosure, delay treatment initiation and worsen adherence [31]. On the other hand, intersectionality highlighted the protective factors such as social support, religious beliefs, personal resilience and altruism as some positive adaptive coping strategies from this population. The positive effects of this shared identity could be protective against undesirable effects from individual and structural stigma and applied to improve health seeking behaviours and outcomes [31,34].

Being sensitive to HIV-related stigma may inform implementers in correctional facilities on ways to deliver HIV care that provide greater privacy. These findings have shown that supportive relationships with staff and other inmates could improve privacy, avoid confidentiality concerns and augment trust [39]. We agree with findings from Kemnitz et al (2017) where recommendations were made to foster relationships that improve social support structures and encourage engagement in HIV care and treatment [40].

Our study had several strengths. The mixed-methods approach provided confidence and validated the findings drawn from the study. Findings from this approach add to existing knowledge gaps. This is likely to improve our understanding of how intersectional stigma affects healthcare seeking behaviour and subsequently design appropriate HIV-related stigma reduction interventions from the positive aspects of shared identities for this population. The purposive sampling of participants at different levels of engagement in HIV care provided representation of the level of care. ART uptake was assessed over 6 months and qualitative findings provided insights into the continuity of care across correctional facilities in South Africa. The recruitment of participants from high and low volume facilities with varied human resource capabilities and HIV prevalence was representative of correctional facilities in South Africa.

Our study had some important limitations. The measures of stigma in the survey instrument were not validated for this population and sample size was too small to enable robust assessment of stigma among our study participants and directly make comparisons to previously published studies. The study findings are not generalizable to all correctional facilities in South Africa and should be interpreted with caution.

## Conclusions

Our study describes ART use and stigma related to HIV status in correctional facilities Involuntary HIV positive status disclosure was a key facet of ART distribution in correctional facilities that led to anticipated HIV-related stigma from incarcerated people not living with HIV. Providing parallel education campaigns to this population will likely lead to destigmatization. HIV stigma-reduction interventions should target all stakeholders in these settings including corrections staff. Programs that promote social interaction among those incarcerated, education on inaccurate HIV beliefs and supportive relationships with corrections staff are also encouraged. Lastly, the development of data collection instruments to better characterize the effects of intersectional stigma on health seeking behaviour is necessary.

## Supporting information

**S1 Checklist. Consolidated criteria for reporting qualitative studies (COREQ): 32-item checklist.**
(DOCX)

**S1 File. Questionnaire.**
(PDF)

**S2 File. In -depth interview guide.**
(PDF)

## Acknowledgments

The authors thank the study participants and staff of the correctional services.

## Author Contributions

**Conceptualization:** Candice M. Chetty-Makkan.

**Data curation:** Pretty Ndini, Israel Rabothata, Danielle Daniels-Felix.

**Formal analysis:** Lucy Chimoyi.

**Funding acquisition:** Christopher J. Hoffmann, Salome Charalambous.

**Investigation:** Harry Hausler, Abraham J. Olivier, Katherine Fielding, Salome Charalambous, Candice M. Chetty-Makkan.

**Methodology:** Christopher J. Hoffmann, Candice M. Chetty-Makkan.

**Project administration:** Pretty Ndini, Danielle Daniels-Felix.

**Supervision:** Katherine Fielding, Candice M. Chetty-Makkan.

**Writing – original draft:** Lucy Chimoyi.

**Writing – review & editing:** Lucy Chimoyi, Christopher J. Hoffmann, Harry Hausler, Abraham J. Olivier, Katherine Fielding, Salome Charalambous, Candice M. Chetty-Makkan.

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
