## [Decision Letter · Decision Letter 0]

17 May 2021

PONE-D-20-29104

HIV-related stigma and uptake of antiretroviral treatment among incarcerated individuals living with HIV/AIDS in South African correctional settings: A mixed methods analysis

PLOS ONE

Dear Dr. Chimoyi,

Thank you for submitting your manuscript to PLOS ONE. After careful consideration, we feel that it has merit but does not fully meet PLOS ONE’s publication criteria as it currently stands. Therefore, we invite you to submit a revised version of the manuscript that addresses the points raised during the review process.

You will find my comments below, along with the feedback from the reviewers. In order to meet the PLOS ONE publication criteria, all of the methodological and reporting concerns must be addressed.

We look forward to receiving your revised manuscript.

Kind regards,

Andrea Knittel

Academic Editor

PLOS ONE

Additional Editor Comments:

In addition to the comments from the reviewers, please consider the following suggestions:

Person-first language is used inconsistently. You may also wish to consider whether the term "correctional" is the best term for jail, prison, and detention facilities. Many in the field have transitioned to using "criminal legal system" and labeling the specific facility type (e.g., "prisons" or "jails and prisons") rather than the collective noun "correctional facilities." If this change is adopted, the addition of a short paragraph in the introduction or methods may be helpful to clarify terminology.

How were the translations checked for accuracy? Were they back-translated or verified in another way?

The methods state that quantitative data were compared using Chi-squared and Fisher's exact test, but these results are not included in the text or the tables.

In describing the qualitative findings, please take care that the phrasing distinguishes between conclusions or statements by the authors and observations from the interviews. As the reviewers highlighted below, this is a critical distinction. Importantly, conclusions or statements from the authors should be moved to the discussion section and only the interview findings should be included in the results section.

There are many small grammatical errors throughout the manuscript. It would benefit from close editing, perhaps by a professional editor.

Journal Requirements:

[The authors thank the study participants and staff of the correctional services. This work was supported by funding from the U.K. Department for International Development (DFID)/ UKAID under grant MMM/EHPDA/AURUM/05150013. The contents of this manuscript are the sole responsibility of the authors and do not reflect the views of DFID, UKAID, or the United Kingdom government. We wish to acknowledge the support from the International AIDS Vaccine Initiative and University of California, San Francisco’s International Traineeships in AIDS Prevention Studies (ITAPS), U.S. NIMH, R25MH0647]

 [SC received the award

U.K. Department for International Development (DFID)/ UKAID under grant MMM/EHPDA/AURUM/05150013

https://www.gov.uk/government/organisations/department-for-international-development

The funders had no role in study design, data collection and analysis, decision to publish, or preparation of the manuscript]

Reviewers' comments:

Reviewer's Responses to Questions

**Comments to the Author**

1. Is the manuscript technically sound, and do the data support the conclusions?

Reviewer #1: Partly

Reviewer #2: Yes

2. Has the statistical analysis been performed appropriately and rigorously? 

Reviewer #1: I Don't Know

Reviewer #2: Yes

3. Have the authors made all data underlying the findings in their manuscript fully available?

Reviewer #1: Yes

Reviewer #2: Yes

4. Is the manuscript presented in an intelligible fashion and written in standard English?

Reviewer #1: Yes

Reviewer #2: Yes

5. Review Comments to the Author

Reviewer #1: This manuscript is a cross sectional study of 219 persons with HIV that used mixed methods to evaluate persons with HIV who were incarcerated to assess effect of stigma on ART uptake. The Parent study was conducted between 9/2016-3/2018 in 3 correctional facilities in South Africa . The majority reported HIV stigma (87%) , and self-coping strategies improved engagement in care. Overall the subject matter is important and the interest in stigma as potentially affecting the HIV treatment care cascade to stay in line with Undetectable=Untransmittable ( U=U) process and treatment as prevention. The paper is well written but I had some questions and comments as outlined below.

Abstract:

Please change to person first language, instead of “HIV positive” , change to persons living with HIV ; instead of “ inmates” – persons in prison/ jail….

Results: of the 219 , n=27 reported no stigma and of them 81% had started ART and of the 192 who reported HIV related stigma 91.75 started ART. Those who reported they were comfortable with HIV status 93.4 % had started ART compared with those not comfortable with diagnosis 83.6% had started ART – not clear if this statistically significant in the text? ( please report P value and CI).

Table 1 and 2 missing is P values.

Hard to know what is meaningful statistically?

The ‘Correctional system structural factors and inadvertent disclosure’ section was highly concerning.

Line 250 . the statement that “ the correctional services environment neither promotes confidentiality of HIV diagnosis nor allows for privacy when receiving HIV services.” is this a theme that emerged or a declarative statement by the authors? It is not clear. IN general this is not the same in all correctional settings- There should be confidentiality- other correctional sites do HIV testing and discuss treatment in other settings in confidential manners.

The fact that they authors state this justifies “ involuntary disclosure” as “unavoidable” in lines 251-252 is not correct either. I can’t tell if this is what happened in the research or a theme they are saying is what the participants reported? If this is a theme then can they reframe these sentences to state what came out or what the participants reported?

For the Qualitative data , it is not clear to me how many of the participants reported these themes? Can they show them in a table format? Number reported ?

Discussion:

2nd paragraph “ the structures put in place by correctional health systems to provide HIV care and ….. inadvertently lead to disclosure of HIV status” – sounds like the authors are saying this as a declarative statement of all correctional health systems – which is not true. I am not clear how they state this for small subject number for the qualitative piece?

I am not sure this is generalizable to all correctional settings.. the authors should discuss this as a limitation .

Reviewer #2: Summary:

This is a cross-sectional study employing mixed methods to investigate stigma among incarcerated people living with HIV in South Africa. The study’s key finding is that although ART use was high, which is positive, 87% of inmates experienced stigma related to their HIV status, which is very alarming and has important implications for an already-stigmatised population due to their incarceration status. Although this is a cross-sectional study from which no inference can be drawn, the study’s strength is its use of mixed methods to explore, in-depth, the experience of stigma among an understudied population in a setting where the HIV epidemic and associated stigma is rampant. It is my view that with the below suggestions for improvement, this manuscript would be suitable for publication.

Abstract:

The authors state in their background that they are investigating the “influence of HIV-related stigma on ART uptake” but this does not seem to be the case. Rather, the authors should consider framing their study as an investigation of the prevalence and experience of stigma among incarcerated people living with HIV.

Introduction:

Lines 85-86: See above comment. With uptake of ART having been high and almost all participants who initiated ART also experiencing stigma, it does not appear to be the objective of this study to look at a relational impact of stigma on uptake. Furthermore, there were no quantitative tests of association conducted between HIV-related stigma and ART uptake, which signals to me that this was not the study’s objective. Authors must consider reframing this aim to fit with the Methods, which appear to indicate that authors researched:1) the uptake of ART; 2) the prevalence of HIV-related stigma among ART users; and 3) the experience of HIV-related stigma among incarcerated people living with HIV.

Methods:

Lines 93-94: Authors state that they assessed “associations” with ART uptake, but nowhere in the manuscript is there a statistical test of association comparing those using ART against those not initiating ART. I believe this is a question of how the authors have framed their methods, but this needs to be clarified by reframing this statement to explain that authors descriptively compared people using ART vs not using ART, otherwise authors should conduct a statistical test of association to show factors associated with ART uptake.

Line 112: Authors should explicitly state that data for their study came from the 6-month time point of the larger TaSP study to make it clear at which time point their cross-sectional study took place.

Lines 147-148: Was there any scoring used to determine if HIV-related stigma took place? What did the authors do with after they combined the six HIV-related stigma questions?

Results:

Authors state in the Methods that they conducted chi-2 and Fisher’s exact tests to compare demographics, stima measures and psychosocial characteristics, but they do not present any P-values for their comparisons of ART users and non-users. Could the authors include P-values from their statistical tests in the text and/or tables?

Figure 1:

It appears from this figure that nurses, and HIV counselling and testing staff also formed part of the interview sample. Is that the case? If it is, I would suggest only reporting incarcerated participants living with HIV as your sample. The N=30 is misleading if it includes people whose views are not represented in the results section.

Discussion:

I think it is important for the authors to mention that although stigma was highly prevalent, it did not appear to impact ART use, as 92% of inmates who reported HIV-related stigma initiated ART.

Lines 360-364: Here the authors state their key findings but do not comment on the intersectionality of individual and structural stigma observed in their study. Since intersectionality was a key theme touched on in the introduction and represented in the survey, it would be good for the authors to dedicate some of the discussion to how intersectional stigma played out in their study.

Authors also briefly mention self-coping mechanisms but do not discuss them any further. Could authors discuss the significance or potential implications of the self-coping mechanisms identified in their study? Could the self-coping mechanisms. Identified in this study be leveraged to develop HIV-related stigma reduction interventions for incarcerated people living with HIV?

Conclusion:

The authors mention that their paper “describes HIV status disclosure” but this is not coherent with the previous sections. The paper described ART use and stigma related to HIV status. Please make this clarification. Authors could also consider adding that involuntary disclosure was a key facet of ART distribution in correctional facilities that might be addressed by providing parallel education campaigns to incarcerated people not living with HIV around destigmatisation.

6. PLOS authors have the option to publish the peer review history of their article (what does this mean?). If published, this will include your full peer review and any attached files.

Reviewer #1: No

Reviewer #2: No

---

## [Author Response · Author response to Decision Letter 0]

1 Jul 2021

Response to comments by Reviewer #1

Abstract:

7. Please change to person first language, instead of “HIV positive”, change to persons living with HIV; instead of “inmates” – persons in prison/ jail….

All references to HIV-positive inmates have been changed to incarcerated people living with HIV. As per ethical guidelines from the South African Department of Correctional facilities, the use of prison or jail is not preferred.

8. Results: of the 219, n=27 reported no stigma and of them 81% had started ART and of the 192 who reported HIV related stigma 91.75% started ART. Those who reported they were comfortable with HIV status 93.4 % had started ART compared with those not comfortable with diagnosis 83.6% had started ART – not clear if this statistically significant in the text? (Please report P value and CI).

The results have been revised to present proportions and not to compare the different groups (on ART vs. Not on ART). The revisions are seen in lines 239, 241 and 243. “Of those comfortable with HIV status, 128/137 (93.4%) had started ART whereas of those not comfortable with HIV status, 56/67 (83.6%) had started ART (Table 2). Of those reluctant to access ART, 33/40 (82.5%) had started ART and of those not reluctant to access ART, 157/170 (92.4%) had started ART. Of those who thought that it was important to keep their HIV status a secret 83/87 (95.4%) had started ART and from those who did not think it was important to keep their HIV status a secret, 105/120 (87.5%) had started ART."

9. Table 1 and 2 missing is P values.

Hard to know what is meaningful statistically? 

The p-values have been included in tables 1 and 2 although the objective of this study was not to assess differences between those who started ART vs those who did not start. We have mentioned this in the methods section (Lines 181-182)

10. The ‘Correctional system structural factors and inadvertent disclosure’ section was highly concerning.

Line 250. The statement that “the correctional services environment neither promotes confidentiality of HIV diagnosis nor allows for privacy when receiving HIV services.” is this a theme that emerged or a declarative statement by the authors? It is not clear. In general this is not the same in all correctional settings- There should be confidentiality- other correctional sites do HIV testing and discuss treatment in other settings in confidential manners. 

We thank the reviewers for this observation. The reference to inadvertent disclosure has been replaced with the emerging theme “fear of inadvertent HIV positive status disclosure while accessing care in “correctional facilities” rather than the declarative statement that was initially made (Lines 270-272).

11. The fact that the authors state this justifies “involuntary disclosure” as “unavoidable” in lines 251-252 is not correct either. I can’t tell if this is what happened in the research or a theme they are saying is what the participants reported? If this is a theme then can they reframe these sentences to state what came out or what the participants reported? 

The sentences (275-276) have been reframed to show that what came out of the participant interviews a fear of disclosure of HIV status during health seeking. The sentence has been revised as follows: “Due to the lack of privacy in a congregate setting, majority of the participants reported fear of HIV status disclosure.”

12. For the Qualitative data, it is not clear to me how many of the participants reported these themes? Can they show them in a table format? Number reported?

A table showing the number of participants reporting these different themes is included in the revised manuscript (Table 3) (Lines 265-268).

Discussion:

13. 2nd paragraph “the structures put in place by correctional health systems to provide HIV care and …..Inadvertently lead to disclosure of HIV status” – sounds like the authors are saying this as a declarative statement of all correctional health systems – which is not true. I am not clear how they state this for small subject number for the qualitative piece? I am not sure this is generalizable to all correctional settings. The authors should discuss this as a limitation.

We thank the reviewer for this observation. Indeed we do not want to make such declarative statements from such a small sample size. This has been stated as a major limitation for this study that limits generalizability of this finding to other correctional facilities in South Africa (Lines 473-474).

“The study findings are not generalizable to all correctional facilities in South Africa and should be interpreted with caution.”

Response to comments by Reviewer #2

Abstract:

14. The authors state in their background that they are investigating the “influence of HIV-related stigma on ART uptake” but this does not seem to be the case. Rather, the authors should consider framing their study as an investigation of the prevalence and experience of stigma among incarcerated people living with HIV.

We thank the reviewer for this observation. We have framed the study to investigate the prevalence and experience of stigma among incarcerated people living with HIV (Lines 31-34). 

“We investigated the prevalence and experience of stigma among incarcerated people living with HIV (PLHIV) in selected South African correctional settings during roll-out of universal test and treat.”

Introduction:

15. Lines 85-86: See above comment. With uptake of ART having been high and almost all participants who initiated ART also experiencing stigma, it does not appear to be the objective of this study to look at a relational impact of stigma on uptake. Furthermore, there were no quantitative tests of association conducted between HIV-related stigma and ART uptake, which signals to me that this was not the study’s objective. Authors must consider reframing this aim to fit with the Methods, which appear to indicate that authors researched:1) the uptake of ART; 2) the prevalence of HIV-related stigma among ART users; and 3) the experience of HIV-related stigma among incarcerated people living with HIV.

We thank the reviewer for this observation. The manuscript has been revised according to this recommendation. We have included these objectives in the introduction (Lines 91-93). We revised the information in the study design to include these objectives (Lines 101-104).

“…describe ART uptake and prevalence of HIV-related stigma among ART users. Then, we qualitatively explored the experiences of HIV-related stigma among incarcerated people living with HIV.”

Methods:

16. Lines 93-94: Authors state that they assessed “associations” with ART uptake, but nowhere in the manuscript is there a statistical test of association comparing those using ART against those not initiating ART. I believe this is a question of how the authors have framed their methods, but this needs to be clarified by reframing this statement to explain that authors descriptively compared people using ART vs not using ART, otherwise authors should conduct a statistical test of association to show factors associated with ART uptake. 

We have taken note of this observation. The objective of the study was not to assess any associations. We did not set out to test for any differences between these two groups (On ART vs. Not on ART). This discrepancy has been corrected as shown in lines 181-182.

17. Line 112: Authors should explicitly state that data for their study came from the 6-month time point of the larger TasP study to make it clear at which time point their cross-sectional study took place. 

Thank you for this observation, this statement “Our study utilized data collected six months post enrolment” has been explicitly indicated in the methods section (Lines 123-124).

18. Lines 147-148: Was there any scoring used to determine if HIV-related stigma took place? What did the authors do with after they combined the six HIV-related stigma questions?

The HIV-related stigma was scored and the resultant variable dichotomized into high and low HIV-related stigma. This has been explained in the methods section (Lines 159-164).

“Four of these questions assessed HIV-related stigma, two focussed on reluctance to seek treatment and the last two on stigma associated with incarceration. Participants choose “Yes (1)”, “No (0)”, or “No answer (3)” and responses with “No answer” set to missing before combining. A scoring ≥1 indicated stigma as previously used and a numerical dichotomized variable with “Yes” and “No” categories was created.”

Results: 

19. Authors state in the Methods that they conducted chi-2 and Fisher’s exact tests to compare demographics, stigma measures and psychosocial characteristics, but they do not present any P-values for their comparisons of ART users and non-users. Could the authors include P-values from their statistical tests in the text and/or tables?

The initial approach was to make the comparisons between the two groups but the co-authors recommended presenting the proportions instead. We have revised the methods to align with the results presented. P-values have been presented. However, no differences have been reported.

Figure 1:

20. It appears from this figure that nurses, and HIV counselling and testing staff also formed part of the interview sample. Is that the case? If it is, I would suggest only reporting incarcerated participants living with HIV as your sample. The N=39 is misleading if it includes people whose views are not represented in the results section.

Thank you for this observation. The figure has been revised to remove the misleading information. The nurses and HCT staff were approached as possible sources for recruitment and were not part of the qualitative study participants.

Discussion: 

21. I think it is important for the authors to mention that although stigma was highly prevalent, it did not appear to impact ART use, as 92% of inmates who reported HIV-related stigma initiated ART. 

Thank you for this observation. This finding has been mentioned in the first paragraph in the discussion in line 404-405. “Participants in this study demonstrated high uptake of antiretroviral treatment (92%) despite a high reported prevalence of HIV-related stigma.”

Lines 360-364: Here the authors state their key findings but do not comment on the intersectionality of individual and structural stigma observed in their study. Since intersectionality was a key theme touched on in the introduction and represented in the survey, it would be good for the authors to dedicate some of the discussion to how intersectional stigma played out in their study. 

Thank you for this observation. Statements on intersectionality of individual and structural stigma have been included from lines 438-441

“Our study findings highlighted intersectional stigma occurring as a result of the convergence of being incarcerated while living with HIV. Our findings showed the negative consequences of individual and structural stigma on incarcerated PLHIV which worsened health seeking behaviours and outcomes”.

22. Authors also briefly mention self-coping mechanisms but do not discuss them any further. Could authors discuss the significance or potential implications of the self-coping mechanisms identified in their study? Could the self-coping mechanisms. Identified in this study be leveraged to develop HIV-related stigma reduction interventions for incarcerated people living with HIV?

Thank you for this observation. Statements on self-coping strategies have been included from lines 441-448.

“Denial of HIV positive status and social exclusion from healthcare services and peers were reported to decrease HIV disclosure, delay treatment initiation and worsen adherence [31]. On the other hand, this intersectional phenomenon highlighted the protective factors such as social support, religious beliefs, personal resilience and altruism as some positive adaptive coping strategies from this population. The positive effects of this shared identity could offer protection against negative effects from individual and structural stigma and could be harnessed to improve health seeking behaviours and outcomes”

 Lines 459 to 463 mention how HIV-related stigma reduction interventions can be designed from the positive aspects of shared identities. 

“Findings from this approach add to existing knowledge gaps. This is likely to improve our understanding of how intersectional stigma affects healthcare seeking behaviour and subsequently design the appropriate HIV- related stigma reduction interventions from the positive aspects of shared identities for this population.”

Conclusion: 

23. The authors mention that their paper “describes HIV status disclosure” but this is not coherent with the previous sections. The paper described ART use and stigma related to HIV status. Please make this clarification. Authors could also consider adding that involuntary disclosure was a key facet of ART distribution in correctional facilities that might be addressed by providing parallel education campaigns to incarcerated people not living with HIV around destigmatisation.

Thank you for this observation. We have made the revision in the conclusion and added the recommendations around parallel education campaigns to incarcerated people not living with HIV (Lines 480-481). 

“Providing parallel education campaigns to this population will likely lead to destigmatization.”

We have included an additional quote under “Perceived stigma in lines 307-315 in the results section to augment the finding that perceived/anticipated stigma was a major deterrent to healthcare seeking in this setting.

“Due to the anticipated stigma, participants rejected services from the peer supporters who assisted healthcare workers in providing care by distributing pre-packaged medication. Most incarcerated PLHIV reported this mode of delivery as one that breached confidentiality of HIV status.

“At the beginning, i said there is no confidentiality. The nurses’ assistants (referring to peer educators) shout out loud during clinic visits that those queuing for the tablets (ART) must step forward to collect. Now they are saying that when I am sick that I should report to the assistant! Where is the confidentiality? Isn’t the assistant a monitor?" Female, 40 years, PLHIV.”

---

## [Decision Letter · Decision Letter 1]

8 Jul 2021

HIV-related stigma and uptake of antiretroviral treatment among incarcerated individuals living with HIV/AIDS in South African correctional settings: A mixed methods analysis

PONE-D-20-29104R1

Dear Dr. Chimoyi,

We’re pleased to inform you that your manuscript has been judged scientifically suitable for publication and will be formally accepted for publication once it meets all outstanding technical requirements. Thank you for your thoughtful revisions and responses to the reviewers.

Kind regards,

Andrea Knittel

Academic Editor

PLOS ONE

Additional Editor Comments (optional):

Reviewers' comments:

Reviewer's Responses to Questions

**Comments to the Author**

1. If the authors have adequately addressed your comments raised in a previous round of review and you feel that this manuscript is now acceptable for publication, you may indicate that here to bypass the “Comments to the Author” section, enter your conflict of interest statement in the “Confidential to Editor” section, and submit your "Accept" recommendation.

Reviewer #1: All comments have been addressed

Reviewer #2: All comments have been addressed

2. Is the manuscript technically sound, and do the data support the conclusions?

Reviewer #1: Yes

Reviewer #2: (No Response)

3. Has the statistical analysis been performed appropriately and rigorously? 

Reviewer #1: Yes

Reviewer #2: (No Response)

4. Have the authors made all data underlying the findings in their manuscript fully available?

Reviewer #1: Yes

Reviewer #2: (No Response)

5. Is the manuscript presented in an intelligible fashion and written in standard English?

Reviewer #1: Yes

Reviewer #2: (No Response)

6. Review Comments to the Author

Reviewer #1: The authors have markedly improved this manuscript and it now seems suitable for publication. I think this has value to improve reduction in stigma for persons with HIV in correctional settings.

Reviewer #2: (No Response)

7. PLOS authors have the option to publish the peer review history of their article (what does this mean?). If published, this will include your full peer review and any attached files.

Reviewer #1: No

Reviewer #2: No

---

## [Editor Report · Acceptance letter]

19 Jul 2021

PONE-D-20-29104R1 

HIV-related stigma and uptake of antiretroviral treatment among incarcerated individuals living with HIV/AIDS in South African correctional settings: A mixed methods analysis 

Dear Dr. Chimoyi:

I'm pleased to inform you that your manuscript has been deemed suitable for publication in PLOS ONE. Congratulations! Your manuscript is now with our production department. 

Kind regards, 

on behalf of

Dr. Andrea Knittel 

Academic Editor

PLOS ONE